# Enhanced Understanding of Horse–Human Interactions to Optimize Welfare

**DOI:** 10.3390/ani11051347

**Published:** 2021-05-09

**Authors:** Katrina Merkies, Olivia Franzin

**Affiliations:** 1Department of Animal Biosciences, University of Guelph, Guelph, ON N1G 2W1, Canada; ofranzin@uoguelph.ca; 2Campbell Centre for the Study of Animal Welfare, University of Guelph, Guelph, ON N1G 2W1, Canada

**Keywords:** positive welfare indicators, senses, affective state, emotional transfer, cognition, personality, attachment styles

## Abstract

**Simple Summary:**

Horses are used by humans for sport, companionship, and as a working animal. Understanding how horses perceive and interact with humans can aid in developing positive interaction approaches, thereby enhancing their welfare. It is also important to be able to recognize behavioural indicators of negative affective states, as the failure to do so when working with horses will negatively impact their welfare when working with humans. The objective of this review is to highlight the current literature surrounding horse–human interaction through the horse’s sensory capabilities, cognition, emotional states, emotional transfer, personality, and attachment styles. This review reveals the various ways horses read humans, such as by our odours, posture, and vocal repertoire, all of which affect the horse’s interpretation of us. While it is difficult to truly know how a horse thinks and feels, paying attention to subtle behavioural signals can give us insight on how horses prefer humans to interact with them. As we are continuously growing our knowledge on how horses view humans, it is important for those working with horses for sport, companionship, or as a working animal to be open to evolving and modifying tactics used to create a positive experience for the horse.

**Abstract:**

Horses (*Equus caballus*) have been domesticated for millennia and are regularly utilized for work, sport, and companionship. Enhanced understanding of human–horse interactions can create avenues to optimize their welfare. This review explores the current research surrounding many aspects of human–horse interactions by first highlighting the horse’s sensory capabilities and how they pertain to human interactions. Evidence exists that suggests that horses can read humans in various ways through our body odours, posture, facial expressions, and attentiveness. The literature also suggests that horses are capable of remembering previous experiences when working with humans. The interrelatedness of equine cognition and affective states within the horse’s umwelt is then explored. From there, equine personality and the current literature regarding emotional transfer between humans and horses is examined. Even though horses may be capable of recognizing emotional states in humans, there remains a gap in the literature of whether horses are capable of empathizing with human emotion. The objective of this literature review is to explore aspects of the relationship between humans and horses to better understand the horse’s umwelt and thereby shed new light on potential positive approaches to enhance equine welfare with humans.

## 1. Introduction

Horses have been domesticated since the third millennium BCE [1] and have played a number of roles in their relationship with humans including war, work, transportation, sport, and companionship. The love of horses permeates human culture, evidenced by an online search that can quickly retrieve months’ worth of reading material about how a horse saved a human’s life, how they touched a human’s soul, how they teach us life skills, how they help us find inner strength and peace, and so on. There is also a surfeit of information on how humans view horses. For example, in a survey asking whether people believed horses could feel emotions, participants strongly agreed that horses were capable of experiencing affective states, including fear, pain, boredom, frustration, depression, sadness, jealousy, anger, happiness, and love [2]. Humans believe in a bond with a horse. Regardless of what country a person comes from, how much money they make, their age, gender, discipline, or experience with horses, people regard their bond with a horse in much the same way. The most frequent references to a bond with the horse involve the horse approaching them, vocally greeting them, trusting them in a frightening situation, taking care of them during hardships, and physically touching them [3]. This implies a universal level of understanding of how humans regard the horse. Additionally, while we have learned much about horse perception, cognition, and emotion, it remains elusive for us to truly know what a horse thinks or feels. Nevertheless, this review will endeavour to explore how horses view humans from the foundation of what science tells us and provide suggestions on how we can work together with our horses to ensure well-being for both parties.

Jakob von Uexküll, a German biologist, coined the term “umwelt” early in the 20th century [4]. Umwelt refers to the perception of one’s environment by one’s own senses and nervous system. Von Uexküll believed that organisms create and shape their own umwelt, guided by their unique evolutionary histories and individual experiences. Thus, even if two animals of the same species reside in the same environment, they can each experience it quite differently. It becomes a challenge for us humans to step outside ourselves to imagine the equine umwelt [5], but that does not mean that we should not try. Accepting horses as sentient beings implies that we accept that they may experience many of the same feelings that we do. Animals have a view that matters, and in our quest to provide a good life for them, we are obliged to consider this point of view [6].

It is important to start with a definition of welfare. Academics may refer to affective states, biological functioning, and natural living to different degrees as welfare, whereas non-academics rely more on their concept of the environment and/or functioning regardless of how the animal may feel. A Delphi study showed that while experts could agree on scenarios where physical welfare was clearly reduced, there was far more discrepancy among them regarding scenarios in which social and emotional needs might be compromised [7]. 

While identifying deficiencies in physical, social, and emotional needs informs us of the welfare status of an animal, more emphasis is now being placed on identifying positive aspects indicative of welfare [8]. When the balance of positive experiences outweighs the negative, then we can consider the animal as having “a good life” [9]. The key is in being able to identify positive indicators of welfare, particularly in human–animal relationships. Suggested measures remain somewhat abstract (e.g., autonomy, positive affect), and still depend on the recognition and reduction of negative aspects [10]. The following review will highlight some aspects of equine perception, cognition, personality, and affective states to better understand how we can endeavour to interact with horses in positive ways. 

## 2. Equine Vision 

We are familiar with the idea that horses can see almost all the way around them, with a blind spot directly in front and behind them. The broad lateral view is enabled by the convex cornea paired with a horizontal visual strip [11,12], and despite not being able to see a full 360° field of view when motionless, the horse need only move his head slightly to achieve this [11]. We also know that horses have monocular vision except for the area directly in front of them. Because the eye is situated on the side of the skull sporting a long nose, binocular vision is limited to a narrow arc of 65–80° [11]. This narrow binocular field of view allows the horse to judge distance, for which the horse needs to raise its head. Horses likely see much better at a distance than humans do, but lack the ability for fine focus on objects near them. While a horse’s vision is inferior to humans in daylight conditions, horses do see better than we do in low-light conditions. In terms of colours, horses appear to be able to distinguish orange, yellow, blue, and green, but have difficulty discerning the colour red [12]. Recent research focused on the colours on jump course obstacles found that white or blue ground lines and fluorescent yellow midrails increased visibility to the horse, resulting in more successful jumping efforts [13]. Early research by Gardner (cited in [14]) found that when signals for food were placed above eye level, horses were less successful on a basic operant task than when stimuli were placed at or below eye level. Research by Hall et al. [14] found that when stimuli were placed at ground level, horses were even more successful. Different from the human retina, the equine retina has no central fovea, a central pit composed of tightly packed cones within the retina [15]. Instead, horses have a “visual strip” which allows the horse to broadly see the entire horizon [15]. To bring an object into focus, a horse will lift, lower, or tilt their head to use their visual strip; therefore, head and neck positions are important factors that affect the visual capability of the horse [15]. 

Equipment or practices that limit or restrict the movement of the horse’s head impact how the horse visually perceives his environment. A simple positive approach is to allow free movement of the head and neck to enable the horse to see and process environmental cues, thereby providing autonomy in how the horse chooses to respond to them. Attention to the colours in a horse’s environment, particularly when asked to navigate coloured obstacles, can not only allow better discernment of obstacles, but also impact rider safety.

## 3. Equine Olfaction 

Horses possess large nostrils and a vomeronasal organ allowing for easy detection and processing of odours. Odours are drawn into the nasal cavity and pass the olfactory epithelium located in the olfactory turbinates before being sensorily processed in the olfactory bulb [13]. The vomonaseral organ is sensitive to non-volatile and poorly-volatile molecules, which are associated with body odours [13]. When an odour of interest is detected, the horse will perform the flehmen response to force air directly into the vomonaseral organ for quick analysis [13]. Despite this knowledge, little research has been done on the impact of odour on communication in horses [16]. Horses engage in sniffing behaviour to recognize kin [16] and this may be why early horse trainers sought to have the horse accept them by rubbing the horse’s chestnuts on their hands and equipment [17]. By smelling conspecifics’ feces or genital areas, horses can detect their state of arousal, health, age, sex, and reproductive status [18,19]. 

However, the horse’s olfactory senses have an even greater capacity than just inferring information from conspecifics: they also have the potential to smell human emotion. In a study done by Sabiniewicz et al. [20], human body samples were collected on cosmetic pads placed under the participants’ armpits in both fear and happiness conditions and then subsequently presented individually to the horses. When smelling fear odour samples, horses raised their head displaying vigilance, and tended to touch a familiar person more often and longer than the unfamiliar research assistant. When smelling the happiness odour, horses lowered their heads and were more relaxed. While some practitioners believe that horses can intuit the emotional state of a person, it could be that the horse is merely responding to olfactory signals. 

Equine olfactory capabilities are deserving of more research to better understand their impact on horse–human interactions. As horses are capable of smelling and understanding our emotional odours, this could influence the horse’s own actions and emotions [21]. 

## 4. Equine Hearing and Vocalization 

Horses do not have a large vocal repertoire. However, they can transmit information and social intentions through vocalizations. Horse vocalizations can convey the sex, size, and reproductive status of the caller [22]. Emotion and arousal are individually encoded within their acoustic qualities [23]. Understanding of a horse’s ability to recognize human vocalizations and their associated intentions would provide valuable information on the cognitive abilities of the species. Research done by Lampe and Andre [24] used a cross-modal expectancy violation setup whereby horses were patted by a familiar human, and subsequently heard a voice recording from either that person or an unfamiliar person. The horses responded faster, gazed longer, and looked more often when hearing voice recordings from the unfamiliar person, demonstrating increased curiosity in the incongruent auditory cues. These findings suggest that horses can identify a familiar human from visual or auditory cues even if one is present without the other. 

This is further supported by work done by Merkies et al. [25], which showed that draft horses, when hearing a recording of a stern voice, increased their heart rate and moved quicker in a round pen, whereas their speed was reduced and they presented their body towards the human more often when a pleasant voice was played. Similarly, horses “froze” longer and held ears forward when hearing human growling recordings compared to human laughter, suggesting vigilance behaviour when presented with aggressive human vocalizations [26]. Conversely, a handler using soothing vocal cues did not enhance the ability of horses to cross a tarpaulin laid on the ground as no differences in task completion, heart rate, and the time to reach calmness were observed compared to using a harsh voice as a reprimand whenever the horse stepped backward [27]. However, within this study, the researchers were examining task completion and not the behaviours associated with how the horses perceived the vocal cues. Thus, the provision of negative reinforcement to learn the task was more salient than the vocal cues. 

Recent novel research examined human use of pet-directed speech toward horses. Pet-directed speech is likened to baby talk, characterized by a higher pitch, slower tempo, and repetition of words. Horses displayed calmer behaviour, more mutual grooming, and were more successful at locating food rewards when humans used pet-directed speech compared to adult-directed speech [28]. This could be simply due to drawing more attention to the person, or, on a higher cognitive level, transference of positive emotion from human to horse, or even the conference of understanding of the human’s intentions. 

Gaining insight on how our vocal repertoire influences the horse’s perception of humans can allow us to work in a way that is beneficial to the horse. Moreover, recognizing how the tone of our voice effects the horse and paying attention to their responses allows for increased human safety when working with horses. 

## 5. Horses in Human Presence 

Horses appear to be better at perceiving human actions than humans are at perceiving theirs. This concept is observed when humans try to approach a horse in the field. For example, approaching a horse in a straight line with direct eye contact while swinging a lead rope causes the horse to move farther away, whereas an indirect approach reduces their flight distance [29]. Wariness of humans appears to be an innate behaviour, as the horses used in this study had minimal handling. The horse’s perception of human action is sensitized if handling is introduced to the horse during their early phases of life. Handling foals immediately after birth (imprint training) results in regressed learners when compared to handling just after weaning [30]. The social environment of the horse will also impact their voluntary interactions with humans. Horses housed in individual stalls away from their conspecifics approach humans sooner, are more easily approached by a human, and express more explorative behavior, less vocalization, and restless behavior, compared to group-housed horses [31]. 

Domestication itself has aided in the development of the horse’s ability to recognize hetero-specific emotions. Through reading human emotional cues, horses can anticipate events and prepare situation-appropriate behaviours. Research from Smith et al. [32] revealed that when horses were shown a photograph of an angry human, they displayed an increased heart rate and left eye gaze bias, indicating the stimuli was perceived negatively. The study of lateralization biases can aid in identifying the horse’s perspective on desired and less desired procedures of training and handling which can, as a result, lead to more efficient training by riders and handlers. Horses trained with positive reinforcement did not show a gaze bias, while ponies trained with negative reinforcement showed a left eye gaze bias toward the human approaching them [33]. Left eye gaze has been repeatedly associated with fear-inducing situations, while right eye gaze has been commonly associated with more positive events [34]. Thus, it may be the quality of work rather than the work itself that may impact laterality.

Horses can not only recognize and understand facial expressions by humans, they can remember these emotional expressions. Research conducted by Proops et al. [35] determined that horses meeting a live person with a neutral face hours after being shown a photograph of the same human displaying either a happy or angry face showed an initial left eye gaze bias. The horses looked longer when meeting the photographed angry person and displayed displacement behaviours such as scratching, sniffing, licking, and chewing. However, there was no difference in horse heart rate when meeting the neutral person regardless of which emotion the horse had viewed in the photograph previously [35]. A study by Merkies et al. [36] examined how horses responded to pairs of humans that visually appeared and moved similarly, however, one of the pair suffered from PTSD while the other did not. While the horses responded similarly to each human behaviourally, they demonstrated a higher heart rate in the presence of the PTSD person. This evidence further suggests that horses are capable of distinguishing between humans. The first report of horses able to discriminate emotions in live humans demonstrated that ponies looked more often with their left eye first towards an angry or sad human compared to a joyful human expression [37]. Baba et al. [38] used a gaze-following task to explore whether horses were sensitive to different human emotional cues. Horses decreased the frequency in which they followed the experimenter’s gaze and the total looking time when the experimenter displayed disgust compared to the neutral condition. Facial expressions are important signals for social bonding, aggression avoidance, navigation of future encounters with specific individuals, and may influence the individual’s own affective state.

Human body posture is an important cue as it can elicit different equine behaviors: for example, round-penning techniques utilize human body posture to produce a desired response from a horse. Evidence by Merkies et al. [39] demonstrated that horses in a round pen moved at a faster gait when the handler faced the horse’s head and moved slower when the handler faced the horse’s hind end. Human attentiveness may also be a factor in how horses view us, as Proops and McComb [40] showed that horses chose to approach an attentive person over an inattentive person. Within this study, it was also observed that the horses would walk around to face the inattentive person, or nudge the inattentive person to gain their attention.

The way in which humans present communicative cues will also affect how horses interpret and understand them. Independent studies done by Maros et al. [41] and Proops and McComb [40] displayed similar results which demonstrated that horses can use information given by a human to locate a hidden reward. Maros et al. [41] performed a two-way object-choice task by hiding food under one of two bowls presented to 20 horses. The horses were successful at locating the hidden reward when the experimenter would point toward the baited bucket continuously until the horse made the correct choice; the horses failed when the experimenter did not sustain the pointing cue. Proops and McComb’s [36] use of a novel marker to indicate the food reward was successful in gaining the horse’s attention, however, tapping, body orientation, and gaze alteration cues from the human did not lead the horses to make the correct choice. It appears that horses may utilize stimulus enhancement to make a choice, as the outstretched hand or the novel marker indirectly attracted the horse to the correct choice.

A horse’s attention can be highly focused to the point that the horse can be completely oblivious to other environmental cues or stimuli. This is known as inattentional blindness. Gabor et al. [42] explored the inattentional blindness phenomenon in horses and found that horses were less reactive during a startle test when their focus was on a food reward at the end of the corridor.

Horses remember past interactions with humans and these experiences impact how the horse will interact with humans in the future. To determine the significance of humans in the horse’s umwelt, Furiex et al. [43] tested school horses with a familiar and unfamiliar person through a variety of approach and handling tests. Different memories formed by horses during previous interactions with humans led them to correspondingly attribute a generally positive or negative significance to humans, and, concerningly, negative experiences had a greater impact on the significance of a human than positive experiences. Moreover, human contact is important to the human–horse relationship, as eliminating human interactions entirely will quickly return horses to feral conditions [44].

Many horse owners wish to believe that their horse loves them more than they love other people. To corroborate this, a study done by Stone [45] demonstrated that when presented with a choice, privately-owned horses spent more time on the side of the pen with their owner than with an unfamiliar person. Hausberger and Muller [46] found that horses managed by the same caretaker responded similarly to the appearance of an unfamiliar person, indicating that differences in individual horse response could be attributed to human factors. The results of this study also showed breed differences, as French Saddlebreds more often displayed friendly behaviour than Anglo-Arabs, whereas Thoroughbreds were more frequently observed to be indifferent to the unfamiliar human. However, Henry et al. [47] found no difference in response of a horse to an unfamiliar person. Evidence by Merkies et al. [36] suggests that horses seek human attention when they are alone, and horses moved slower, carried their head lower, vocalized, and chewed less when any human was with them compared to when they were left alone. Even though humans may be important to a horse, recent evidence shows that horses perform a target task better when rewarded with food compared to a human touch reward such as scratching and petting [48].

Horses read humans in various ways, such as through our body posture, facial expressions, and attentiveness. Small actions such as a happy facial expression when approaching the horse can allow for a positive experience with the horse. As horses remember previous experiences, more positive experiences with humans will promote generalization to all human interactions. Recognizing how the horse responds to a positive interpretation of a human can create better welfare when working with humans.

## 6. Equine Affective or Emotional States

The horse’s face is a great source of information (Figure 1). The pain face in a horse is characterized by stiffly backward ears, tightening around the eyes, tension above the eye area, prominent chewing muscles, tense muzzle, pronounced chin, and flared nostrils [49]. Behavioural indicators of stress include wide eyes, ears back, nostrils flared, head tossing, high head carriage, trembling, and clamped tail [50]. Conversely, the pleasure face of a horse is commonly described with relaxed eyes, extended upper lips, and loosely backward ears. However, behavioral and physiological markers of discrete positive emotions remain scarcely investigated in horses. To determine new emotional markers in horses, Lansade et al. [51] used tactile stimulations to introduce physical sensations of contrasting valence. Horses who were gently groomed were more often observed with their necks moderately raised, eyes half-closed, and their upper lips extended and either immobile or twitching, while their ears were pointed backward closely in line with their nose. In contrast, when horses received a more vigorous form of grooming they were more frequently observed with their neck in a high position, eyes wide open or showing the whites, contracted lips with the corner of the mouth raised jerkily, and asymmetrical ears.

Another approach used by researchers to examine how a horse feels about a certain experimental factor is simply asking the horse through the use of a preference test. For example, Lee et al. [52] showed that a horse will choose to go to a paddock with food and company over an empty paddock or work on a treadmill. Horses distinctly preferred being ridden with a natural head position compared to a hyperflexed position [53]. Mejdell et al. [54] showed horses had a clear preference for wearing blankets in freezing temperatures.

Behavioural indicators of stress such as rearing, bolting, kicking, or biting may be obvious to handlers, however, even if such indicators are absent, this does not translate to the absence of distress within the horse. Moreover, undesirable behaviours can lead to escalating punitive actions by the handler and as a result, with each successive interaction, the horse will learn to suppress its natural response. The aim of a trainer is to habituate horses to not respond to particular stimuli, such as billowing flags or judge’s booths, but the training approach is critical. Gradual desensitization allows a horse to become accustomed to a low level of an aversive stimulus before increasing the intensity. Flooding occurs when the horse is presented with the full intensity of an aversive stimulus while being prevented from escaping until he no longer resists [55]. Flooding can easily slip over into learned helplessness, which occurs when the animal learns that no matter what behaviour they perform, they cannot escape the aversive stimulus and ultimately give up trying to avoid it. Learned helplessness results in a lack of motivation, a compromised ability to learn new tasks, and depression [56].

Thus, there is a fine line between training a horse for a specific task, and training them to mask the expression of their underlying “undesirable” emotion. Human failure to recognize behavioural signs of distress or misinterpretation of these signs can cause implications for horse welfare as well as horse and human safety. A study done by Bell et al. [57] found a discrepancy in the ability of experts and lay survey respondents to recognize negative affect in horses. Lay respondents were more likely to categorize natural horsemanship and bridleless riding scenarios as positive; however, they were unable to identify negative behaviours associated with these scenarios. Survey respondents who indicated they train mainly using positive reinforcement (i.e., clicker trainers) demonstrated a better understanding of behaviour, as they were able to correctly identify negative affect more often than the other respondents. Concerningly, some respondents indicated that they would treat their own horse similarly even though distress was evident. Even experts could not reach agreement on the level of stress a horse was experiencing during veterinary care [50].

It is important to recognize behavioural indicators of negative affective states, as our failure to recognize these when working with horses will adversely affect the horse’s immediate and future welfare. However, we must also educate ourselves on behavioural indicators of positive affective states and attempt to increase their frequency.

## 7. Equine Cognition

An animal’s emotion and their welfare are interrelated [34], as negative emotion will drive an animal to a solution while positive emotion will stimulate an animal to seek it more. For example, thirst will drive a horse to find water, while having a preferred companion will stimulate a horse to reunite with them. Emotions are a large component of how an individual perceives an event, as they shape an individual’s umwelt. From this interrelatedness, emotions can play a key role in cognition through the subjective perception of a current situation. This relationship between emotion and cognition can be explored through a cognitive bias test. Henry et al. [58] found that gentle contact with humans accompanied by more natural living conditions were associated with a more optimistic horse.

The environment is another factor that can affect a horse’s cognition. Horses housed in groups is a basic indicator of positive welfare, being that social interaction is such a high priority for horses [59]. Social isolation impairs cognitive ability, as documented in shelter dogs housed alone who showed decreased cognitive ability when asked to choose between a large and small plate of offered food compared to dogs housed in groups or living with people [60]. Similarly, horses kept on pasture with other horses learned quicker, displayed fewer undesirable behaviours, and required less desensitization time to novel equipment [61].There is also a genetic component to learning linked to the emotionality of the horse, as temperament traits are linked to specific breeds [62]. Learning ability has been shown to differ among breeds, with similar heritability coefficients for both learning and emotionality [34].

It is difficult to clarify these interrelationships between emotion and cognition, as individuals will vary. Brubaker and Udell [63] examined cognition and learning in horses, highlighting the importance of recognizing the emotionality of the horse during training. For example, less reactive horses will perform better at a negative reinforcement task such as learning to move away from leg pressure, while reactive horses perform better at an avoidance task such as crossing an obstacle to avoid a puff of air [64,65]. Moreover, reinforcement, when performed correctly, can be a beneficial training tool, however, if used incorrectly, can be associated with increased stress. Positive reinforcement techniques generate higher motivation from the horse to participate and engage in more exploratory and anticipatory behavior. Horses learn better and faster with a greater positive attitude toward their trainers and fewer problem behaviours displayed within their training sessions when positive reinforcement is used [66]. When a task is first learned via a positive experience, this leads to subsequent facilitation of learning and human–horse interactions. For example, horses taught a simple verbal “stay” command through positive reinforcement were able to generalize this command to different contexts, such as veterinary care and basic handling procedures, and they engaged in more affiliative behaviours with a familiar and unfamiliar human during training [33].

The cognitive abilities of a horse can be quantified by relationships built from repeated interactions and previous experiences with humans. The creation of a relationship implies a high level of social cognitive ability, as this is an important attribute to acquire within social species like horses. Sankey [33] demonstrated that horses were capable of discriminating between positive and negative interactions. As cumulative encounters will define the relationship, this generalized to positive interactions with any humans interacting with the horses even months later. Similar long-term results were reported by Hanggi and Ingersoll [67] who showed that horses were able to discriminate between previously trained rewarding and non-rewarding three-dimensional household objects as well as two-dimensional photographs of those objects even up to seven years after the initial training. Friendliness toward humans is a clear positive indicator associated with improved welfare in working horses [60].

The horse’s previous experiences working with humans impacts their cognitive abilities. As humans play a large role in equine cognition, it is important to understand how our interactions can impact this. We must take pains to ensure we are training in positive scenarios and pay attention to both the positive and negative behavioural indicators of stress.

## 8. Emotional Transfer between Horses and Humans

Dogs are sensitive to not only cues from conspecifics but also to human emotional cues [68], suggesting that domestication may influence such sensitivity. Different results have been documented regarding the horse’s interpretation of human emotional cues. Earliest studies have shown that individuals stroking horses with negative thoughts induced an increase in horse heart rate [69]. Additionally, handlers with positive attitudes could lead a horse through a course with greater ease [70]. Emotional state may be influenced by the person interacting with the horse, as horses are more willing to approach people dependent on the attitude of their caretaker [46,71]. Recent research determined that horses watching a video showing an unfamiliar horse and human interacting in either a positive or negative scenario responded congruently both behaviourally and physiologically, suggesting emotional contagion from either the horse or the human [72].

Common belief in the equine community is that horses will sense their rider’s nervousness and become nervous in turn, which will impact their overall performance [73], but this notion has been refuted by other researchers. Through the examination of cortisol release, heart rate, and heart rate variability in the horse and the rider, von Lewinski et al. [74] found that the presence of spectators caused more pronounced changes of cardiac activity in the riders than it did in their horses during a performance. Similarly, Merkies et al. [25] showed that horses in the presence of humans who were fearful of them did not display elevated heart rates or an increase in stress-induced behaviours compared to when in the presence of calm humans.

Another common belief is that horses can intuit emotional distress in humans and will act altruistically to alleviate the human’s distress. However, this belief may credit the horse with more empathy, and thus higher cognitive level, than they truly possess. Work done by Merkies et al. [25,36] demonstrated that behavioral and physiological responses of horses to humans were more pronounced based on human experience with horses rather than whether the human experienced mental distress. Moreover, these studies also provided evidence that deliberate body movements by those with acquired horse experience make the horse more attentive to them.

While horses may recognize different emotional states in humans, there is a gap in knowledge whether the horse is capable of empathizing or deliberately acting to alleviate specific human emotions. The synchronization of heart rates is an alluring idea indicating shared sympathetic experiences between individuals, however, the synchronization of heart rates between horses and humans has not been definitively proven [75,76]. If the emotional state of a human has the potential to influence the emotional state of the horse, it is important for us to work with a positive attitude based on our knowledge of equine affective states.

## 9. Equine Personality and Attachment Styles

Equine temperament is genetically predetermined, with the transmittance of traits from generation to generation allowing for selective breeding resulting in distinct differences between breeds of horses. For example, Thoroughbreds and Arabians rank high in traits like anxiousness, excitability, and nervousness while breeds such as the American Quarter Horse and draft breeds rank low [77]. A review of equine personality by Rankins and Wickens [78] outlined multiple temperament traits and dimensions that exist in horses, as personality is genetically based and modified by the horse’s environment and previous experiences. Equine personality is of interest to behaviour and welfare scientists as well as equine practitioners, as certain traits can impact injury risk, training processes, learning, and the hors–human relationship. Personality is entrained based on environment and experience during the critical periods of development in which an animal’s behaviour and personality are more easily modified by external events such as socialization [78]. Exposing a fearful horse to a novel environment will decrease the learning capacities of the horse, as fearful horses are more likely to switch from goal-directed behaviours to a habitual process or routine [51].

It is clear that humans become very attached to their horses, however, it is not as clear if horses are equally attached to their human. Attachment is commonly defined as a reciprocal bond that develops over time. There are four main pillars associated with attachment styles: proximity seeking, secure base, safe haven, and separation distress [79]. These have been characterized in some species, including humans and dogs, but are less well-defined in horses. Horses can display proximity-seeking, as horses with a single handler more readily approached humans than those with multiple handlers. Henry et al. [58] demonstrated that a foal will retreat to its mother as a secure base and horses are more willing to explore when accompanied by a conspecific; however, there is no clear evidence of a secure base between a horse and a human. It has been observed that horses will approach a novel object more readily when led by a human, perhaps relying on the human as their “safe haven”. However, Ijichi et al. [80] found no difference in proactive or reactive behaviour of the horse, eye temperature, or heart rate variability when horses were handled by the owner or an unfamiliar person during a stressful scenario, indicating that a safe haven can be provided by any human and is not reliant on a bond. Horses do show distress when alone, indicating separation anxiety, however, there is no literature relating to their distress when separated from a human as long as other horses are present. A recent study by Lundberg et al. [81] specifically examined all four pillars of attachment in a separation-reunion experiment with horses. Their results were similar to previous studies, showing that horses do display proximity seeking and a safe haven effect, but these were independent of the human involved.

Understanding the specific personality traits can allow humans to modify how they work with their horse, with the aim of increasing positive experiences. Further research needs to be undertaken regarding attachment styles to gain a better understanding of the bond between a horse and human. While we may provide some comfort by our presence, it still remains to be determined if a reciprocal bond exists.

## 10. Conclusions

Interactions between humans and horses are comprised of many different aspects evolving from the various perspectives of this historical relationship. Understanding the workings of this relationship can enable advancements in welfare, training, husbandry, and management, particularly through focusing on indicators of positive experiences for the horse. Within this literature review, it is evident that some aspects of the human–horse relationship are generally well understood, such as equine perception; however, other aspects of this relationship, such as the emotional transfer between humans and horses, are inconclusive and require greater scientific attention. Whether the relationship we have with our horse is as a working animal, for sport, or companionship, working toward optimizing this relationship is essential to improve equine welfare when working with humans.

Understanding how horses interpret humans is fundamental to understanding their sensory capabilities. Knowledge concerning equine vocalizations allows insight into the affective state of a horse, and could be extrapolated to the way horses react to the different repertoire of human vocalizations. Understanding how our tone of voice affects the horse will allow for increased human safety when working with horses. This can aid in creating more effective bilateral communication between humans and horses. Moreover, through examining horses in the presence of humans, our body posture, attentiveness, and transmission of communicative cues are all key assets to creating and maintaining a positive relationship with the horse. Equine affective or emotional states can be influenced by our interactions and attitudes around them. As equine emotion and cognition are interrelated, they are integral components to the comprehension of a horse’s umwelt. As we humans become increasingly attached to our horses, it is important to understand equine personality and their true attachment to humans. Knowing that our interactions with the horse are cumulative, keeping our interactions positive will result in the horse maintaining a better outlook on humans. Horses are a relevant component to society and work closely with humans as companions, therapy animals, and team members. Recognizing the ways we can improve our relationship with them will allow for better welfare of horses living in the human world. It is important for us to educate ourselves to better understand behavioural indicators of positive affective states and strive to increase their frequency of use.

## Figures and Tables

**Figure 1 animals-11-01347-f001:**
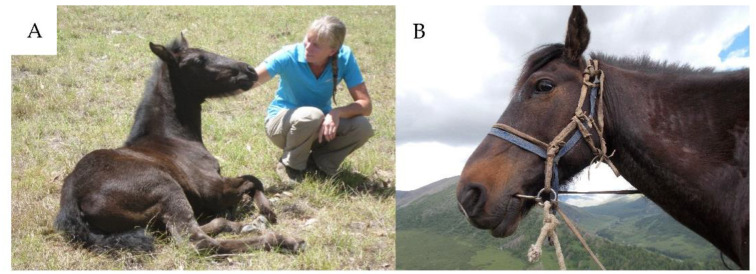
The equine face can reveal much about the animal’s affective state. (**A**): A young horse with what could be described as a “pleasure” face. Note the relaxed eyes, extended upper lips, and loosely backward ears (Photo credit P McGreevy). (**B**): A horse displaying indications of stress including tightening around eyes, tension above the eye area, pronounced chin, and flared nostrils (Photo credit K Merkies).

## Data Availability

Not applicable

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
