# Peer review of "Enhanced Understanding of Horse–Human Interactions to Optimize Welfare"

_animals, 2021, doi:10.3390/ani11051347_

Round 1

Reviewer 1 Report

The paper is well-written and of interest to a wide audience.

Apart from a couple of minor details, the only thing I suggest the authors reconsider is the title and their reliance on the Churchill quote.

I do not believe the quote is appropriate or useful here, albeit rather clever to interpret in this way.

Lines 41-2: really take ownership of the sentiment? What exactly is the sentiment Churchill was trying to convey and how is it relevant to your review?

Lines 48-9: repetition of 'quick' - suggest rewording

Lines 62-5: Really? Churchill's saying has preserved the horse's place in society? Are we only looking at the 'outside' of a man? Surely, particularly when you discuss human emotions and physiological parameters, you are indeed discussing the 'inside' of the human?

Line 73: Animals have a view that matters.... Suggest you reword this for clarity

Line 77: non-specialist (folk)? what do you mean here? Non-academics, pleasure riders/owners? 

Line 339: 'drive the animal to a solution' - reward? This needs more explanation

Line 361: I suggest you give an example of an 'avoidance task' as readers will likely understand negative reinforcement but not this

Author Response

We thank all the reviewers for their useful and supportive comments. We have responded to each comment as below.

The paper is well-written and of interest to a wide audience.

Apart from a couple of minor details, the only thing I suggest the authors reconsider is the title and their reliance on the Churchill quote.

I do not believe the quote is appropriate or useful here, albeit rather clever to interpret in this way.

Lines 41-2: really take ownership of the sentiment? What exactly is the sentiment Churchill was trying to convey and how is it relevant to your review?

We have taken this reviewer’s suggestions and retitled the paper and reworked the opening paragraph.

Lines 48-9: repetition of 'quick' - suggest rewording

reworded

Lines 62-5: Really? Churchill's saying has preserved the horse's place in society? Are we only looking at the 'outside' of a man? Surely, particularly when you discuss human emotions and physiological parameters, you are indeed discussing the 'inside' of the human?

We have deleted this sentence and reworded the end of this paragraph.

Line 73: Animals have a view that matters.... Suggest you reword this for clarity

done, and added reference 6

Line 77: non-specialist (folk)? what do you mean here? Non-academics, pleasure riders/owners? 

Reworded as non-academics

Line 339: 'drive the animal to a solution' - reward? This needs more explanation

We have included examples L378-379

Line 361: I suggest you give an example of an 'avoidance task' as readers will likely understand negative reinforcement but not this

Done – added examples for both types of tasks for clarity L400-403

Reviewer 2 Report

This is a beautiful written and well considered paper bringing together a breath of research into a comprehensive review. I would like to congratulate the authors, it has been a real pleasure to review this paper.

I would be happy to recommend publication of this paper as it stands. My only very minor comment is that I don't think you need to specify that the human was live in Line 216.

Author Response

We thank all the reviewers for their useful and supportive comments. We have responded to each comment as below.

I would be happy to recommend publication of this paper as it stands. My only very minor comment is that I don't think you need to specify that the human was live in Line 216.

Many thanks for the gracious comments. We prefer to leave the word “live” in this context as to distinguish from previous research that had used photographs of humans.

Reviewer 3 Report

The Outside of a Horse – What it means for Horse and Human

Thank you for allowing me to review this paper.  It is a timely review in the field of equine wellbeing and I consider this to be a valuable addition to literature in this (and wider) academic areas.

General Comments;

  • Well structured and well written with a logical flow to the review content
  • This is a useful review for applied animal behaviour work in the equine world
  • Very readable and interesting

Simple Summary

  • Clear and appropriate with acknowledgement of limitations of knowing what an animal thinks and feels

Abstract

  • Accurate precis of the paper
  • Clear, concise, precise
  • Appropriate keywords

Introduction

  • Ideal coverage and introduction – coverage of key terms and definitions to set the scene

Equine Vision

  • Query whether worth noting this - If horses had toes: demonstrating mirror self recognition at group level in Equus caballus | SpringerLink

Equine Olfaction

  • Appropriate coverage of human and horse odour recognition

Equine Hearing and Vocalisation

  • Query whether worth noting this study about equine sensitivity to ‘baby talk’ - Horses are sensitive to baby talk: pet-directed speech facilitates communication with humans in a pointing task and during grooming | SpringerLink – I feel this is really interesting in terms of horse-human interactions

Horses in human presence

  • Line 176-177 Are horses better at recognising human actions via learning or is it innate? Could this be a wider point of note? (I think this is really fascinating!) – domestication is mentioned later however.

Equine Affective or Emotional States

  • Line 320 – this is a really important finding in terms of human-horse interactions; human awareness and experience linked to welfare perceptions
  • No specific mention of ‘learned helplessness’ in equines? – a particular issue in therapy horses and riding school mounts - Is There Evidence of Learned Helplessness in Horses?: Journal of Applied Animal Welfare Science: Vol 11, No 3 (tandfonline.com) I am aware of the ongoing debate here!

Equine Cognition

  • I really like this section with applied value identified also!

Emotional Transfer between horses and humans

  • TYPO - BETWEEN IS SPELLED BRETWEEN – line 393

Equine Personality and Attachment Styles

  • Appropriate review of these areas

Conclusion

  • Appropriate conclusion
  • Highlights additional areas for further research based on the review
  • Suggested applied value of findings/review outcomes

References

  • Appears fine – I have not however finely gone through and proof read every reference!
  • Good level of consideration of literature
  • No excessive self-referencing that is unjustified

Author Response

We thank all the reviewers for their useful and supportive comments. We have responded to each comment as below.

Equine Vision

  • Query whether worth noting this -If horses had toes: demonstrating mirror self recognition at group level in Equus caballus | SpringerLink

this is interesting new research, but as our paper deals specifically with horses in human presence, there does not appear to be an appropriate place to include this information. We feel that it is reflective of horses’ cognition, but as humans were expressly absent in the research on mirror recognition, it does not precisely fit into this review.

Equine Hearing and Vocalisation

  • Query whether worth noting this study about equine sensitivity to ‘baby talk’ - Horses are sensitive to baby talk: pet-directed speech facilitates communication with humans in a pointing task and during grooming | SpringerLink – I feel this is really interesting in terms of horse-human interactions

thank you for bringing this recent paper to our attention. It is certainly well-placed in our review.

We have included this information in this section L181-188

Horses in human presence

  • Line 176-177 Are horses better at recognising human actions via learning or is it innate? Could this be a wider point of note? (I think this is really fascinating!) – domestication is mentioned later however.

good point, and we have included a brief mention to this innate behaviour L199-200

Equine Affective or Emotional States

  • Line 320 – this is a really important finding in terms of human-horse interactions; human awareness and experience linked to welfare perceptions
  • No specific mention of ‘learned helplessness’ in equines? – a particular issue in therapy horses and riding school mounts - Is There Evidence of Learned Helplessness in Horses?: Journal of Applied Animal Welfare Science: Vol 11, No 3 (tandfonline.com) I am aware of the ongoing debate here!

excellent point.

We have included a paragraph on flooding and learned helplessness L346-354

Emotional Transfer between horses and humans

  • TYPO - BETWEEN IS SPELLED BRETWEEN – line 393

fixed

Reviewer 4 Report

Dear Authors,

attached please find a pdf with comments.

Regards

Author Response

We thank all the reviewers for their useful and supportive comments. We have responded to each comment as below.

L22 – include reference

we prefer not to put a reference in the abstract so have altered the wording to be more general. However we have included this information appropriately referenced in the opening paragraph.

L97 - I encourage you to enter more accurate information at this point (for example: the horse's eyeballs are convex and directed laterally which provides a wide field of view. The horse can see almost all around (within 330 degrees) with little head movement, but at the cost of limiting the binocular field (to 65 degrees), which is contributed to by the elongated shape of the facial part of the head, which creates a blind spot in front.

Added information to the first paragraph in this section.

L101 - Please enter also information how the horse reacts to the color white. Is it easily recognizable by the horse and does it improve the visibility of the obstacle, e.g. at riding events?

We included information from Paul and Stevens recent study on colours on jump obstacles.

L121 - Here I also encourage you to include, for potential readers, an anatomical diagram of the horse's nostrils and the location of Jakobson's organ. Please also include information which areas of the brain do pheromones affect? Information has been added.

We decided not to include a diagram as there is an excellent diagram available in Rorvang et al.’s paper.

L289 - A valuable addition would be photographs depicting these equine emotions observed by the authors.

We have added two photos showing the pleasure and stressed face (Fig 1).